# 28-Homobrassinolide Primed Seed Improved Lead Stress Tolerance in *Brassica rapa* L. through Modulation of Physio-Biochemical Attributes and Nutrient Uptake

**DOI:** 10.3390/plants12203528

**Published:** 2023-10-11

**Authors:** Mawra Khan, Shakil Ahmed, Nasim Ahmad Yasin, Rehana Sardar, Muhammad Hussaan, Abdel-Rhman Z. Gaafar, Faish Ullah Haider

**Affiliations:** 1Institute of Botany, University of the Punjab, Lahore 54590, Pakistan; 2SSG RO-II Department, University of the Punjab, Lahore 54590, Pakistan; 3Department of Botany, Faculty of Life Sciences, Government College University, Faisalabad 38000, Pakistan; 4Department of Botany and Microbiology, College of Science, King Saud University, Riyadh 11451, Saudi Arabia; 5South China Botanical Garden, Guangzhou 510520, China

**Keywords:** 2,2-diphenylpicrylhydrazyl (DPPH), lead, seed priming, 28-homobrassinolide

## Abstract

Brassinosteroids (BRs) influence a variety of physiological reactions and alleviate different biotic and abiotic stressors. Turnip seedlings were grown with the goal of further exploring and expanding their function in plants under abiotic stress, particularly under heavy metal toxicity (lead stress). This study’s objective was to ascertain the role of applied 28-homobrassinolide (HBL) in reducing lead (Pb) stress in turnip plants. Turnip seeds treated with 1, 5, and 10 µM HBL and were grown-up in Pb-contaminated soil (300 mg kg^−1^). Lead accumulation reduces biomass, growth attributes, and various biochemical parameters, as well as increasing proline content. Seed germination, root and shoot growth, and gas exchange characteristics were enhanced via HBL treatment. Furthermore, Pb-stressed seedlings had decreased total soluble protein concentrations, photosynthetic pigments, nutrition, and phenol content. Nonetheless, HBL increased chlorophyll *a* and chlorophyll *b* levels in plant, resulting in increased photosynthesis. As a result, seeds treated with HBL2 (5 µM L^−1^) had higher nutritional contents (Mg^+2^, Zn^+2^, Na^+2^, and K^+1^). HBL2-treated seedlings had higher DPPH and metal tolerance indexes. This led to the conclusion that HBL2 effectively reduced Pb toxicity and improved resistance in lead-contaminated soil.

## 1. Introduction

The viability of the agroecosystem is being threatened by the presence of various environmental contaminants, including heavy metals, which has an impact on both human and animal health [1,2,3]. Metal toxicity causes a variety of morphological, biochemical, physiological, and ultrastructural changes in plants [4]. The overaccumulation of these metals (loids) by plants can lower crop quality and output, endangering the safety and security of our food supply [5].

Among the large number of causative agents that will be involved in spreading heavy metal contamination from the environment, the most crucial ones are industries and agricultural fields. Heavy metals accumulate in soil in minimal amounts that are within permissible limits and are not harmful. Other man-made factors, such as industries, mining, and agriculture, have increased the accumulation of heavy metals to toxic levels [6]. Heavy metal toxicity is one of the most promising global threats in today’s fast-paced world. Heavy metal concentrations from various man-made sources exceed permissible limits and reach harmful levels, resulting in the destruction of agricultural soil [7].

The prominent heavy metals are cadmium, chromium, lead, mercury, and arsenic, which are perilous and have no biological function in plants [8]. The unfavorable exposure of heavy metal leads to deleterious effects on plants. It will affect plant yield and other physical and biological parameters, leading to retardation in the growth of plants [9]. The major causes involved in elevating the level of heavy metals are industrialization, urbanization, pesticide application, the use of chemical fertilizers, smelting, mining activities, and the inappropriate management of fossil fuels and wastes [10].

Lead is highly ubiquitous, strong, and toxic in nature, having no crucial role in plant metabolism [3]. One of the riskiest heavy metal contaminants in the environment is lead, originating from a variety of sources such as lead ore mining and smelting, coal combustion, effluents from storage battery industries, automobile exhausts, metal plating and finishing operations, fertilizers, pesticides, and additives in pigments and gasoline [11].

A large number of morphological, biochemical and physiological phenomenon are untowardly affected by the induction of Pb into plants [12]. The growth and metabolic processes of plants are hampered by lead toxicity [13]. Molecular damages are caused by Pb generating reactive oxygen species (ROS) such as hydrogen peroxide (H_2_O_2_), hydroxyl radicals (OHs) and superoxide radicals (O_2_) either directly or indirectly [11,12,14]. The stress-induced changes caused by Pb exposure include declines in the growth rate, reductions in biomass, the chlorosis of plant leaves and other biochemical and physiological changes [15].

Plants undergo a diverse number of biotic, abiotic, and environmental stresses during their entire life cycle including their growth and metabolism [16]. Fluctuating temperatures, drought, and metal toxicity are some of the most prominent abiotic stresses which are manifested in plants. These factors lead to marked and unforeseeable losses in crop productivity in the field of agriculture [17]. In today’s fast-paced world, modern policies are employed, the focus of which is to elevate the per acre yield of harvests and decrease losses post- and pre-harvest [18]. Several metabolic, physiological and biochemical changes are correlated with drought or salinity stresses [19]. They will affect plant activity, the rate of metabolism of plants and in turn yield via oxidative stresses [20].

For harmonizing the growth rate of single seeds, seed priming is employed [21]. For lessening the emergence time, ensure uniformity in appearance, and provide the betterment of crop standards in horticultural and field crops, improved seed renewing techniques are applied [22]. 

A class of naturally occurring plant hormones known as brassinosteroids has been identified to have an impact on a number of physiological processes in plants [23]. Additionally, they are crucial in reducing a variety of biotic and abiotic stressors [24]. The exogenous administration of brassinosteroids improved resistance to moisture, drought, and low-/high-temperature environments. Similar to how the BR treatment reduced salt stress in plants, Nunez et al. [25] found that it also improved the antioxidant system in plants under salinity stress (reviewed by Clouse and Sasse [26], and Sasse [23]).

The purpose of this study was to investigate and elaborate on the beneficial role of 28-homobrassinosteroid in *Brassica rapa* L. subjected to lead (Pb) stress, as well as to determine whether or not a potential link exists between physiochemical properties and the level of BRs’ resilience to stress in turnip seedlings. The hypothesis tested is that 28-homobrassinolide reduces the poisonous effects of lead on the progression of turnip plants by elevating the level of DPPH activity and nutrient uptake.

## 2. Results

### 2.1. Estimation of Growth Parameters and Biomass Yield

Turnip seed germination is hampered by lead. HBL, on the other hand, displayed a remarkable growth rate in Table 1. Turnip plants exposed to Pb stress showed a 38.89% decrease in the germination percentage compared to that of the control group. Turnip seeds treated with HBL2 demonstrated an increase in the germination percentage of 5.5% when compared to that of the control. Turnip seeds under Pb stress supplemented with HBL2 developing plants exhibited an increase of 43% compared to those under the Pb-only treatment.

Turnip saplings exposed to lead stress had lower fresh and dry weights of roots and shoots when compared to those of the controls. Primed seeds of HBL, conversely, enhanced growth characteristics and lessened Pb stress in *B. rapa*. Furthermore, when compared to the control, HBL2 treatments showed 1.6-fold, 1.7-fold, 1.25-fold, and 1.5-fold increases in shoot fresh weight, root fresh weight, shoot length, and root length by, respectively (Table 1). When compared to Pb-only conditions, HBL2 increased shoot length, root length, shoot and root fresh weight by 1.5, 1.3, 3.4, and 3.1 times in lead-spiked soil (Table 1).

### 2.2. Estimation of Gas Exchange Attributes

Pb toxicity negatively affect the rate of photosynthesis (*A*), stomatal conductance (*gs*), and transpiration rate (*E*) of *B. rapa* leaves. In contrast to the control, lead stress reduced net photosynthesis (*A*), stomatal conductance (*gs*), and transpiration rate (*E*) by 47, 72, and 60%, respectively. As compared to control, HBL1 and HBL3 treatment enhanced *A*, *gs*, *E* by 27, 3, 24, 36, 9 and 42%, respectively. In the case of lead-stressed soil, the impact of seeds primed with HBL2 on photosynthetic rate was substantial. Transpiration rate, and stomatal conductance, increasing these parameters by 2.2, 1.5, and 1.9-fold, respectively, under Pb stress conditions, whereas HBL2 showed enhancement in net photosynthesis (*A*), stomatal conductance (*gs*), and transpiration rate (*E*) by 1.6-fold, 1.4-fold and 1.48-fold, respectively in comparison to control (Figure 1).

### 2.3. Estimation of Photosynthetic Pigments and Carotenoids

Pb stress resulted in lower chlorophyll and carotenoids levels in seedlings compared to those in controls (Table 2). HBL application via seed priming increased Chl *a*, Chl *b*, total chlorophyll, and carotenoids, assisting the plants in mitigating Pb toxicity. In comparison to the control, Pb reduced Chl *a* content by 53.4%, Chl *b* content by 32.5%, total chlorophyll content by 38.5%, and carotenoid content by 61%. The use of 28-homobrassinolide increased chlorophyll and carotenoid content. Among other treatments, HBL2 depicts the highest levels of Chl *a*, Chl *b*, total chlorophyll, and carotenoid content. In the case of HBL2, the Chl *a* and Chl *b* content were increased 2.1- and 2.7-fold, respectively, when compared to those of the control. In the case of Pb-spiked soil, HBL2 resulted in an elevation. In the case of HBL2, the Chl *a* and Chl *b* content were increased 2.1- and 2.7-fold, respectively, when matched to the control. HBL2 + Pb showed a 3.8-fold elevation in Chl *a*, 3.9-fold increase in Chl *b*, and a 3.7-fold increase in carotenoids in Pb-spiked soil. HBL1 and HBL3 boosted Chl *a* (58%; 88%) and Chl *b* (1.5-fold; 2.5-fold), and total Chl content by 1.57 and 2.34 times in comparison to those of the control.

### 2.4. Estimation of Heavy Metal and Nutritional Content

Pb consumption was nominal in the HBL2 treatment. In the treatment with HBL2, Pb uptake was minimized by 65%. The values of Pb uptake in HBL1 and HBL3 were 23% and 37.6% lower, respectively. The nutritional content of plants was influenced by lead toxicity (K^+^, Na^+^, Mg^+2^, and Zn^+2^). Plants upraised from seeds primed with HBL, on the other hand, had recovered nutrient contents reflecting that of both control and Pb-affected plants. Table 3 shows that seeds primed with HBL2 had improved nutrient uptake and accumulation in both control and Pb-affected plants. Primed seeds with HBL2 showed increases of Mg^+2^ by 1.3-fold, Zn^+2^ by 1.1-fold, Na^+^ by 1.4-fold, K^+^ by 1.3-fold in comparison to control. Pb-affected plants had the lowest nutrient content.

### 2.5. Assessment of Metal Tolerance Index and Accumulation Coefficient Factor

The metal tolerance index (MTI) in the case of Pb treatment was 28.1 (Table 2). Turnip plants grown from HBL2-treated seeds had an increased MTI that was 5.01 times higher than that of Pb-treated plants. HBL1 + Pb, HBL2 + Pb, and HBL3 + Pb exhibited 94.8, 141, and 124 MTIs, respectively. The maximum accumulation factor (AC) of the Pb-spiked turnip plant was 6.3 (Table 2). The accumulation factor (AC) in HBL1 + Pb, HBL2 + Pb, and HBL3 + Pb was 3.69, 4.007, and 3.17, respectively.

### 2.6. Estimation of Proline Content

Lead toxicity resulted in a 2.3-fold increase in proline content in *B. rapa* when compared to that of control plants (Figure 2). Turnip lants grown from seeds primed with HBL also had increased proline content. In comparison to the HBL2-only treatment, the Pb-affected plants treated with HBL2 had 1.29 times more proline. Pb-affected plants treated with HBL1, HBL2 and HBL3 showed 1.48-fold, 1.53-fold and 1.5-fold enhanced proline content in comparison to that of plants with Pb-only treatment, respectively.

### 2.7. Estimation of Protein Content

Lead-stressed plants showed a decline in protein content (Figure 2). Pb caused a decrease in protein content of 69% compared to that of control plants. In the case of 28-homobrassinolide, HBL2 showed the highest protein content among other treatments. HBL2 resulted in 1.4-fold increases followed by HBL1 (1.32-fold) and HBL3 (1.35-fold) compared to control-treated plants. Among the Pb stress treatments, HBL2 + Pb showed a maximum result of enhanced protein content increasing 1.6-fold in comparison to the Pb-only stress treatment.

### 2.8. Estimation of Total Phenol

Pb stress reduced phenolic content (Figure 2). When compared to control plants, Pb-stressed plants had a 52% decrease in phenolic content. HBL2 produced the best results in the HBL application. In comparison to the control, HBL2 increased phenolic content 1.43 times followed by HBL1 (1.3-fold) and HBL3 (1.41-fold). Among lead-stressed plants with primed HBL seeds, HBL2 + Pb plants had 1.5 times the phenolic content of Pb-stressed plants.

### 2.9. 2,2-Diphenylpicrylhydrazyl 0(DPPH) Free Radical Scavenging Activity Test

Table 3 showed that the Pb stress descries 81% DPPH activity. When associated with the control, HBL2 showed a 1.5-fold increase in %age DPPH. Among Pb-stressed plants, HBL2 had the highest %age DPPH by 1.6-fold when compared to the Pb-only treatment.

### 2.10. Correlation between Various Growth and Physiological Parameters

In Pb-contaminated soil with HBL-primed seeds, a Pearson’s correlation also quantified a relationship with the growth and physiological characteristics of plants (Figure 3). Proline concentration, the metal tolerance index, and accumulation factor are all positively linked with Pb uptake in *B. rapa.* Similarly, Pb absorption is inversely connected with protein content, DPPH scavenging activity, magnesium content, sodium content, zinc content, and potassium content. It is also inversely correlated with shoot length, root length, shoot fresh weight, root fresh weight, shoot dry weight, and root dry weight. A heatmap histogram was also illustrated in this experiment (Figure 4). They showed the same results as those from the correlation.

### 2.11. Principal Component Analysis (PCA)

To show a connection between the morphological and physiological characteristics of *B. rapa* cultivated in Pb-contaminated soil using HBL-primed seeds, principal component analysis (PCA) loading plots were created (Figure 5). The majority of all components are made out of the two primary components, Dim1 and Dim2. Dim1 makes up 79.8% of the entire dataset, whereas Dim2 makes up 11.8%. Protein, DPPH scavenging activity, magnesium, sodium, zinc, and potassium contents, shoot length, root length, shoot fresh weight, root fresh weight, shoot dry weight, and root dry weight are all positively correlated. There was significant negative correlation of variables with proline content, the metal tolerance index, and the accumulation factor in *B. rapa* under Pb stress.

## 3. Material and Methods

*Brassica rapa* L. (turnip) seeds were sterilized by immersing them in a 0.5% sodium hypochlorite solution for 3 min and then thoroughly washing them three times with distilled water. Briefly, 28-homobrassinosteroid was prepared as a 1 mM stock solution, and subsequent dilutions were performed in 500 mL of distilled water using 0.5, 2.5, and 5 mL of the stock solution. The seeds were primed for 12 h in the dark with 1, 5, and 10 µM HBL solutions, which are referred to as HBL1, HBL2, and HBL3, respectively. Primed seeds were properly rinsed before being placed on blotting paper for dehydration at room temperature for 3 h to retain original moisture content. Unprimed seeds were utilized as a control.

The Botanical Garden, University of the Punjab, Lahore, served as the site of the experiment. This site is located in the south of the city (74 21-00-E; 31 35-00-N). The soil (2 kg) in the assigned plastic pots was amended with Pb (NO_3_)_2_ (lead acetate) as a source of Pb (300 mg kg^−1^ Pb). Pb-spiked soil was regarded as a Pb treatment. Pb-contaminated or non-contaminated (control) pots were placed in greenhouse conditions for 15 days prior to seed sowing. For the experiment, thirty-two (32) clay pots of eight (8) treatments (C = Control, Pb = 300 mg kg^−1^, HBL1 = 1 µM, HBL2 = 5 µM, HBL3 = 10 µM, HBL1 + Pb, HBL2 + Pb, and HBL3 + Pb) were arranged in accordance with the randomized complete block design, abbreviated as RCBD. There were four replicates for each treatment. In total, 5 HBL-primes and unprimed turnip seeds were sown in each pot in a completely random order. Thinning was conducted when the plants were mature enough, which was about 25 days after the initiation of germination, and two seedlings were kept in each assigned pot. The germination percentage was calculated by using the following formula: Germinated seeds/Total no. of seeds ×100. Destructive harvesting was carried out 48 days after germination.

### 3.1. Assessment of Growth Parameters and Biomass Yield

The root and shoot parts of harvested *B. rapa* plants were separated for further analysis. Each plant’s shoot length, leaf area, root length, and leaf number were all measured. The length of the root and shoot was measured with a ruler. After analyzing fresh biomass from plant parts, the samples were oven-dried at 70 °C for two days to estimate dry biomass production.

### 3.2. Assessment of Gas Exchange Attributes

Stomatal conductivity (*gs*), transpiration rate (*E*), and current photosynthetic rate (*A*) were measured using an infrared gas analyzer (IRGA). The reading was taken between 9:00 and 10:00 a.m. using a completely stretched turnip leaf and an LCA-4 analyzer system (ADC, Ltd. 12 Spurling works, Pindar Road, Hoddesdon, Herts, UK).

### 3.3. Estimation of Photosynthetic Pigments 

The amounts of total chlorophyll, chlorophyll *a* and chlorophyll *b* were calculated. In 10 mL of 80% acetone, 100 mg of turnip leaf tissues was suspended. The mixture was carefully mixed and left overnight at 4 °C in the dark. The next day, absorbance was measured using a spectrophotometer at 480, 663, and 645 nm (Uv-1800 240V. CAT. No. 206-25400-38 SHIMZDZU Corporation, Kyoto Japan). Ultimately, the amount of chlorophyll *a*, chlorophyll *b*, and consequently the total amount of chlorophyll was examined by using the formula by Arnon given below [27]:$$Chlorophyll‘a’\left(\frac{mg}{g}FW\right)=\left[\frac{\left(0.0127\times A663-0.00269\times A645\right)\times 100}{0.5}\right]\;Chlorophyll‘b’\left(\frac{mg}{g}FW\right)=\left[\frac{\left(0.0229\times A645-0.00468\times A663\right)\times 100}{0.5}\right]\;Total\;Chlorophyll\left(\frac{mg}{g}FW\right)=\left[\frac{\left(0.0202\times A645-0.00802\times A663\right)\times 100}{0.5}\right]$$

### 3.4. Estimation of Carotenoid

The same extract used to assess chlorophyll “*a*” and “*b*” content was used to estimate carotene content in plant leaves. The Kirk and Allen method [28] was used for this purpose. The extract was placed in cuvettes and analyzed spectrophotometrically (Shimadzu UV-1800) using the formula provided by Davies [29].

### 3.5. Estimation of Heavy Metal and Nutritional Content 

The amount of Pb in turnip tissues was measured. After 24 h of oven drying at 70 °C, all samples were completely crushed. A 0.5 g sample was taken and digested with 5 mL (70%) of nitric acid (HNO_3_) and 1.5 mL (60%) of chloric acid (HClO_4_). The solution was heated on a hot plate until the brown fumes disappeared. After cooling the samples, 5 mL of diluted (1:1) HCl (density 1.18 g/mL) was added. Finally, distilled water was added in an amount of up to 25 mL to dilute the solution. The solution was filtered and stored in plastic bottles for further analysis using an atomic absorption spectrophotometer (GBC XPLOR AA-Dual).

The nutritional content of the digested samples was determined using a flame photometer, and the amounts of Mg^+2^, Zn^+2^, Na^+2^, and K^+1^ were determined.

In the case of heavy metal agglomeration in roots and shoots, to assess the plant–soil relationship, the accumulation co-efficient (AC) was calculated using the formula provided by Al-Farraj et al. [30]:$$AC\;Factor=\frac{Concentration\;(Shoot\;or\;Root)}{Concentration\;of\;Soil}$$

C_shoot or root_ = the concentration of heavy metal (Pb) in *B. rapa*.

C_soil_ = the concentration in soil

Furthermore, the metal tolerance index (MTI) was evaluated by using formula given by Balint et al. [31]:$$\%MTI=\frac{Dry\;Weight\;of\;Treated\;Plants}{Dry\;Weight\;of\;Untreated\;Plants}\times 100$$

### 3.6. Estimation of Proline Content

The approach developed by Bates et al. [32] was used to estimate the content of prolines. Briefly, 1 g of the leaf sample was mixed with 10 mL of 3% sulfo-salicylic acid before centrifugation at 11,500× *g*. The leaf extract was then mixed with the same amount of glacial acetic acid, acid ninhydrin, and phosphoric acid. The prepared solution was allowed to sit at 100 °C in a hot water bath (N.S Engineering concern XMTG-9000). After 1 h, 4 mL of toluene was poured, and chromophore was used to extract it. The solution was kept at 25 °C for 0.5 h, and the colorimetric value was set at 520 nm for comparison with the standard curve.

### 3.7. Estimation of Protein Content

According to Lowery et al. [33], the protein content of harvested plants was determined using the 1 g of shredded plant matter collected. Crushed plant matter was centrifuged for 15 min at 6000 rpm in 2 mL of 1N phosphate buffer. After centrifugation, 0.4 mL of the supernatant was collected in an unused test tube. After shaking, 2 mL of the Folin mixture was poured into the tube and allowed to sit at room temperature. Finally, optical density was quantified using a spectrophotometer at 750 nm. The soluble protein of each sample was computed using a standard curve.

### 3.8. Assessment of Total Phenol

Briefly, 2 g of fresh plant material sample was added to 10 mL of aqueous methanol (80%) for 15 min at 65 °C to quantify phenolic content. An amount of 1 mL of this extract was taken, 250 µL of Folin–Ciocalteau reagent (1N) and 5 mL of distilled water were poured, and the mixture was stored at 30 °C. The absorptivity of the reaction mixture was measured at 725 nm and compared to the gallic acid standard curve to determine the total amount of phenol [34].

### 3.9. 2,2-Diphenylpicrylhydrazyl (DPPH) Free Radical Scavenging Activity Test

The DPPH Free Radical Scavenging Activity Test was performed [35]. Briefly, 1 mL of methanolic extract and 5 mL of 0.1 Mm DPPH methanolic solution were mixed thoroughly and kept in the dark for 60 min. Finally, using a spectrophotometer, the absorptivity of the reaction mixture was calculated at 517 nm. A blank was also used and was created by reusing the extract with methanol (1 mL).
$$Scavenging\;Activity(\%)=\left[1-\left(\frac{A517\;nm,Sample}{A517\;nm,Blank}\right)\right]\times 100$$

### 3.10. Statistical Evaluation

Using SPSS v.20 (Chicago, IL, USA) software, the one-way variance of the generated data observations was examined using Duncan’s multiple range test. Values show the means and standard errors of four replicates. Significant differences between the treatments are indicated by non-identical letters when *p* = 0.05 is used. The link between the variables was measured using Pearson’s correlation analysis. Using the RStudio program, the Pearson correlation, heatmap histogram, and results of the principal component analysis (PCA) were evaluated.

## 4. Discussion

Because of the toxicity caused by Pb, root and shoot growth was stunted in terms of length and both fresh and dry mass (Table 1). Pb exposure causes stress-induced changes in plant growth, biomass, the chlorosis of plant leaves, and other biochemical and physiological changes [36]. At least in the early phases of toxicity, it is believed that cell elongation is essentially what prevents root growth, whereas reduced cell division can definitely impair growth in the later stages [37,38]. The effects of repressed root growth and shoot growth were observed (Table 1). Brassinosteroids shield plants from environmental stresses such as salinity [39,40], water [41], temperature [42], moisture [43], and heavy metal (Cd and Ni) stress [44,45].

Pb toxicity disrupts stomatal conductance, transpiration rates, and photosynthetic rates (Figure 1), which results in shortages of nutrients, particularly P, K, Ca, and Mg [46]. 

Furthermore, the lessened photosynthesis, chlorophyll and carotenoid content resulted in a lessening of the progression of Pb-stressed plants (Table 1). However, when the strained plants were treated again with 28-homobrassinosteroids (HBL), the values improved when compared to those of the untreated stressed plants. According to Sharma et al. [47], treating seeds with 28-homobrassinolide improved *B. juncea*’s germination when exposed to nickel stress. The fact that brassinosteroids are known to promote cell growth and division suggests that they affect germination via promoting embryonic development and the megaspore’s function in mitotic divisions [48].

Brassinosteroids elicit such a response due to their participation in the alteration of plasma membrane construction and penetrability under stress conditions [49], in addition to promoting the antioxidative defense system. In addition, BRs increase protein and nucleic acid synthesis [50], activate proton pumps [51], and control gene repression and depression [52], enzyme activities, and photosynthetic pigments [53] (Table 2). Abdullahi et al. [54] showed that the use of brasssinolide enhanced growth and raised the concentration of chlorophyll pigment in mung bean seedlings under aluminum-related stress. 

Chlorophyll and carotenoids were dramatically depleted in the presence of Pb (Table 2). Through the process of photosynthesis, leaves are crucial to the generation of food [55]. We think that BRs increased the production and activation of enzymes involved in both photosynthesis and the production of chlorophyll, which increased the rate of photosynthesis (Figure 1). Such a belief is supported by Yu et al. [56] in their study, which showed that BRs boosted the activity of rubisco, a crucial enzyme in photosynthesis and related activities. These authors postulate that an increase in rubisco activity was responsible for the increased Calvin cycle’s CO_2_ absorption. Brassinosteroid treatment increased chlorophyll content and photosynthetic rates in *B. juncea* under cadmium and nickel stress [45,57]. 

Pb toxicity inhibits mineral uptake and causes disruptions in physiochemical activities, growth parameters, and developmental scales [58]. The electromotive force behind the decrease in mineral nutrient was reduced due to the depolarization of the plasma membrane of various root cells [59]. Lead is known to block several ions from entering their absorption sites on roots [60], thereby obstructing their uptake. Similarly, cucumber, maize, tomato, and lettuce plants had lower Ca, K, and Mg content in Cd-contaminated areas [61]. Zhang et al. [62] presented the comparable results of the decreased uptake of Ca, Mg, N, and P content in Medicago sativa plants that were under the influence of Cd. 

The increased uptake of heavy metals in plants reduced the uptake of essential nutrients such as magnesium, iron, and zinc, resulting in a decrease in chlorophyll production by hundreds of percent [63]. The reduction in chlorophyll could be mediated by a decrease in Mg uptake [46], which is a necessary part of the chlorophyll molecule. Magnesium ion levels appear to be extremely sensitive to Pb toxicity [64]. The augmented Mg content in HBL-treated plants in this study can be attributed to an increase in oxidative phosphorylation, which leads to an increase in chlorophyll synthesis. Zinc also serves as a cofactor of several enzymes involved in plant physiochemical processes [14]. Zn is obligatory for the regulation and maintenance of gene expression in plants for them to be resistant to environmental stresses [65].

Potassium (K^+^) controls the turgor of leaf cells, enhancing photosynthetic absorption, nutrition intake, and leaf inclination. Controlling stomatal opening and enabling appropriate gas and water fluxes depend on potassium [66]. Facilitating a well-structured stroma lamella, increased K^+^ concentration in chloroplasts is also required to support chloroplast integrity and light absorption efficiency. Rubisco production and activity have been found to be severely suppressed by K^+^ shortage while under stress [67]. Beyond its function in ribosomal transport and mobility, K^+^ has other impacts on protein synthesis rates. Thus, K^+^ regulates rubisco activity by influencing biosynthesis, chloroplast stability, and nutrition, water, and gas fluxes [68]. Nutrient absorption was boosted when the Pb-stressed seedlings were given a subsequent HBL treatment (Table 3).

In order to combat the toxicity brought on by stress, which is mediated by the formation of reactive oxygen species such as superoxide hydrogen peroxide (H_2_O_2_) and hydroxyl radical (HO•) radicals [69], increased enzyme activity and proline levels are a general response to various environmental and heavy metal stress. The current study additionally demonstrates that plants exposed to Pb and/or BRs have higher proline levels (Figure 2). The increased proline concentration in the roots may be the result of direct lead exposure. As a general reaction to varied abiotic stressors, plants produce proline. It is also proposed that proline acts as a source of carbon and nitrogen for rapid stress recovery, as well as being a stabilizer of plasma membranes and some macromolecules and a free radical scavenger [70] defending plants from harsh-stress situations.

Protein synthesis can be used to assess plant growth because some major proteins are crucial for plant endurance and protection under stressful circumstances [71]. Lead phytotoxicity reduces plant protein content (Figure 2). This decrease in protein concentration is caused by lead accumulation in the plant, which causes the disintegration of soluble proteins in the cell. Seeds primed with 28-homobrassinolide demonstrated improved metal stress tolerance as well as increased protein levels for cell stability.

Phenolics are one of the promising classes of antioxidants with a distinct feature. Metal ions can be chelated by phenols, which reduces the unpleasant effect of redox-metal ions [72]. The secondary role of phenolics is to prevent the production of reactive oxygen species via stopping the chain reaction involving free radicals [73]. Phenolic content declined under lead stress while the follow-up treatment with HBL enhanced its quantity (Figure 2).

During metabolism, plants synthesize complex secondary metabolites [74]. Some of these metabolites have the ability to eliminate free radicals from plant cells. However, during abiotic stress, the metabolites’ scavenging power declines [75]. Table 3 clearly demonstrates that antioxidant potential in HBL-primed turnip seedlings was enhanced by increased DPPH activity. It has been widely reported that DPPH forms a persistent complex by releasing H atoms, and color intensity determines the capacity of bearing potential to detoxify reactive oxygen species in stressed plants.

## 5. Conclusions

The current study found that using 28-homobrassinosteroid (HBL) treatment increased the growth response to Pb stress. The increased resistance caused by the use of HBL was reflected in improved plant growth, photosynthesis, proline, chlorophyll, and carotenoids under the presence of lead. Current research indicates that turnip plants grown under lead stress show low germination, root growth, shoot growth, and biomass. HBL-treated seedlings, on the other hand, greatly reduced Pb toxicity and restored physiochemical activity, allowing them to grow faster. Plants were able to withstand Pb toxicity due to the increased synthesis of osmoprotectants, phenolic and protein content and DPPH scavenging activity. The current study suggests investigating the efficiency of HBL seed priming before planting agricultural crops in Pb-affected soil for sustainable crop management. Nonetheless, further molecular research will reveal the comprehensive mechanism of HBL in Pb stress reduction and growth enhancement.

## Figures and Tables

**Figure 1 plants-12-03528-f001:**
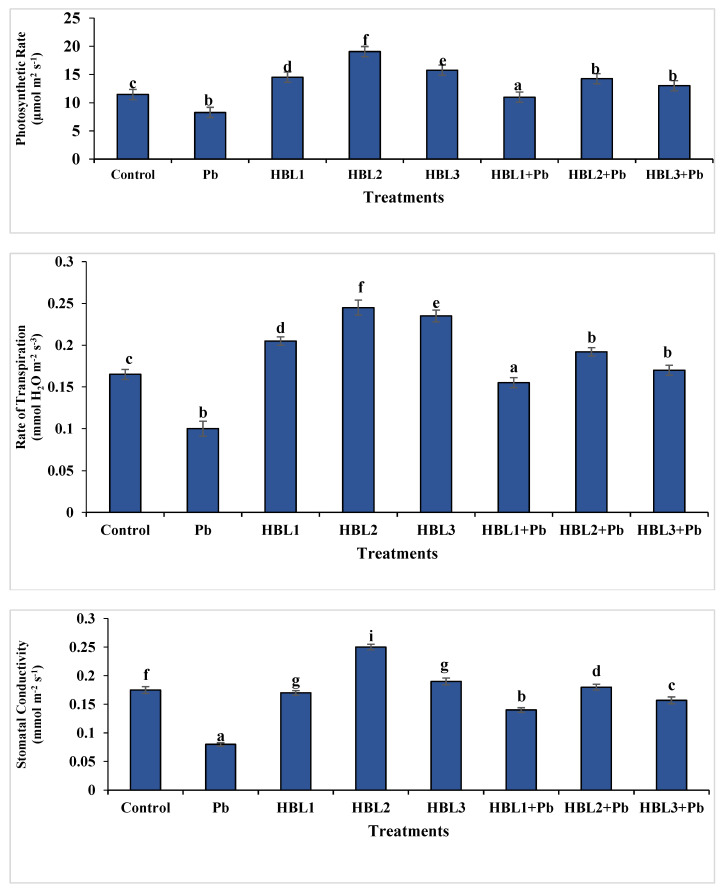
The Effect of 28-homobrassinolide on photosynthetic rate, transpiration rate, and stomatal conductance of *B. rapa* under lead tress. Data exhibit means ± SE of 4 replicates. Non-identical letters indicate significant differences between the treatments at *p* ≤ 0.05. C = control, Pb = 300 mg kg^−1^ Pb, HBL1 = 5 µM HBL, HBL2 = 10 µM HBL, and HBL3 = 15 µM HBL.

**Figure 2 plants-12-03528-f002:**
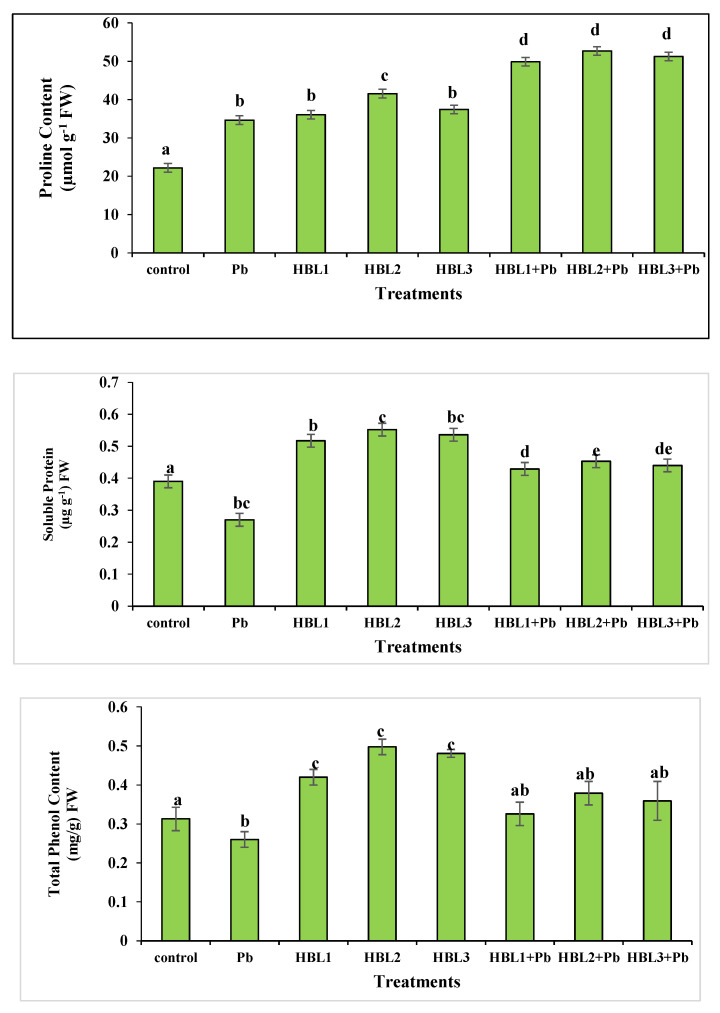
The effect of 28-homobrassinolide on the protein content, total phenolic content, and proline content of *B. rapa* under lead stress. Data exhibit means ± SE of 4 replicates. Non-identical letters indicate significant differences between the treatments at *p* ≤ 0.05. C = control, Pb = 300 mg kg^−1^ Pb, HBL1 = 5 µM HBL, HBL2 = 10 µM HBL, and HBL3 = 15 µM HBL).

**Figure 3 plants-12-03528-f003:**
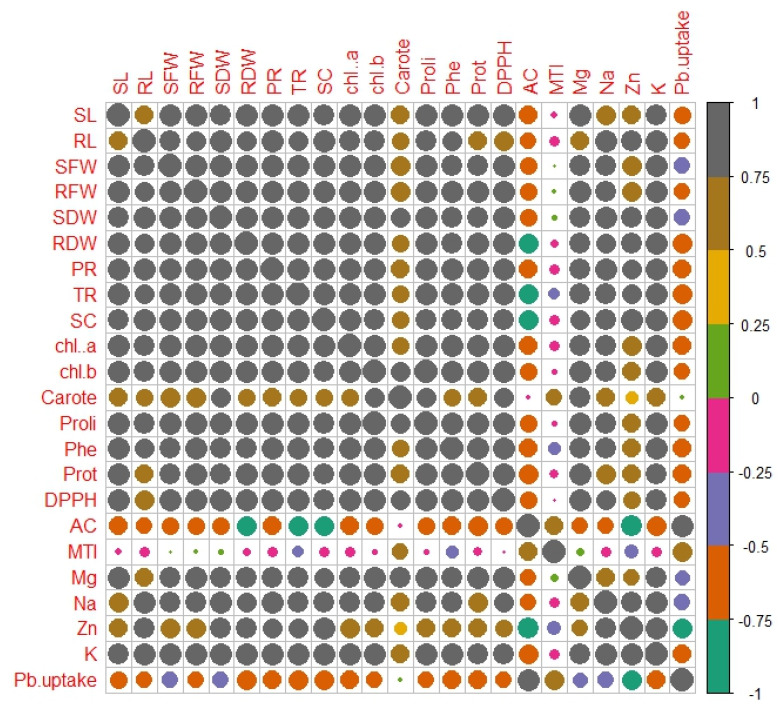
Correlation between different attributes studied in *B. rapa.* Different abbreviations used in the figure are as follows: SL (shoot length), RL (root length), SFW (shoot fresh weight), RFW (root fresh weight), SDW (shoot dry weight), RDW (root dry weight), PR (photosynthetic rate), TR (rate of transpiration), SC (stomatal conductance), Chl *a* (chlorophyll *a*), Chl *b* (chlorophyll *b*), Carote (carotenoid), Proli (proline content), Phe (phenolic content), Prot (protein), DPPH (DPPH scavenging activity), AC (accumulation factor), MTI (metal tolerating index), Mg (magnesium content), Na (sodium content), Zn (zinc content), K (potassium content), and Pb uptake (lead uptake).

**Figure 4 plants-12-03528-f004:**
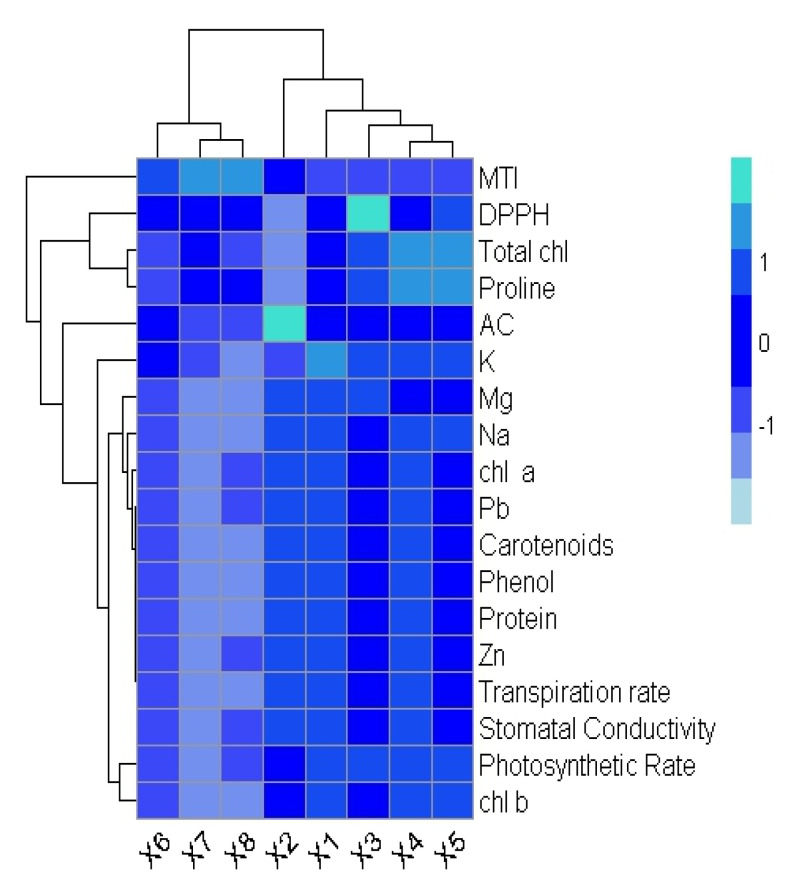
Heatmap histogram correlation between different physiological and biochemical traits of *B. rapa.* The abbreviations are as follows: MTI (metal tolerance index), DPPH (DPPH scavenging activity), Total Chl (total chlorophyll), Proline (proline content), AC (accumulation factor), K (potassium content), Na (sodium content), Mg (magnesium content), Chl *a* (chlorophyll *a*), Pb (lead content), Phenol (phenolic content), Protein (protein content), Zn (zinc content), and Chl *b* (chlorophyll b). V1 = Control, V2 = Pb = 300 mg kg^−1^, V3 = 1 µM, V4 = 5 µM, V5 = 10 µM, V6 = HBL1 + Pb, V7 = HBL2 + Pb, and V8 = HBL3 + Pb.

**Figure 5 plants-12-03528-f005:**
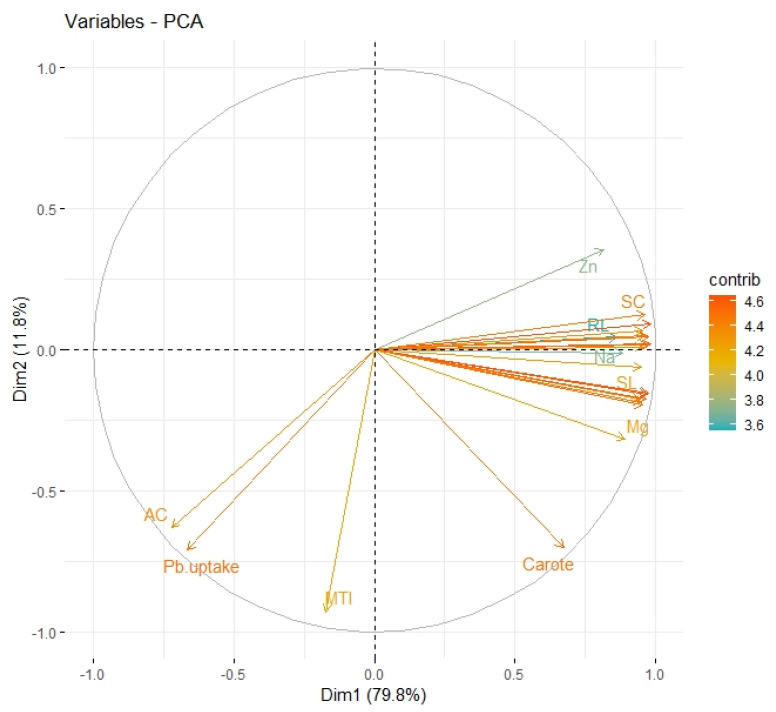
Loading plots of principal component analysis showing a relationship between growth and biochemical parameters of *B. rapa* in lead-contaminated soil. Different abbreviations used in the figures are as follows: AC (accumulation factor), Pb (lead uptake), MTI (metal tolerance index), Carote (carotenoid content), Zn (zinc content), Mg (magnesium content), SL (shoot length), Na (sodium content), RL (root length), and SC (stomatal conductance).

**Table 1 plants-12-03528-t001:** Effect of 28-homobrassinolide on the growth and biomass production of *B. rapa* under lead stress.

Treatments	Growth Traits						
	Shoot Length (cm)	Root Length (cm)	Shoot Fresh Weight (g plant^−1^)	Root Fresh Weight (g plant^−1^)	Shoot Dry Weight (g plant^−1^)	Root Dry Weight (g plant^−1^)	Germination %
C	24.2 ± 0.25 b	11.5 ± 0.06 c	25.8 ± 0.02 c	4.18 ± 0.04 b	2.6 ± 0.04 b	0.38 ± 0.04 c	90 ± 0.14 bcd
Pb	18.75 ± 0.25 a	9.28 ± 0.04 a	8.9 ± 0.02 b	1.43 ± 0.04 a	0.79 ± 0.04 a	0.04 ± 0.004 a	55 ± 0.125 a
HBL1	29.4 ± 0.3 de	11.4 ± 0.07 c	28.5 ± 0.02 cd	5.38 ± 0.05 c	2.68 ± 0.02 c	0.44 ± 0.01 d	92.5 ± 0.185 cd
HBL2	30.49 ± 0.02 e	18.25 ± 0.4 f	43.59 ± 0.02 f	7.21 ± 0.04 f	5.03 ± 0.04 e	0.557 ± 0.07 h	95 ± 0.125 d
HBL3	29.6 ± 0.4 de	12.6 ± 0.4 de	37.73 ± 0.02 d	6.35 ± 0.04 d	3.72 ± 0.04 d	0.483 ± 0.01 f	94.4 ± 0.14 d
HBL1 + Pb	25.12 ± 0.7 bc	10.3 ± 0.08 b	25.7 ± 0.01 a	3.95 ± 0.2 b	2.54 ± 0.04 b	0.307 ± 0.01 b	75 ± 0.125 b
HBL2 + Pb	28.7 ± 0.4 de	12.63 ± 0.1 e	30.56 ± 0.02 e	6.48 ± 0.04 e	3.85 ± 0.04 d	0.417 ± 0.04 g	79 ± 0.135 bcd
HBL3 + Pb	25.5 ± 0.2 e	12 ± 0.4 de	25.78 ± 0.06 cd	5.28 ± 0.2 b	3.34 ± 0.06 c	0.39 ± 0.02 e	76.5 ± 0.185 bc

Data exhibit means ± SE of 4 replicates. Non-identical letters indicate significant differences between the treatments at *p* ≤ 0.05. C = control, Pb = 300 mg kg^−1^ Pb, HBL1 = 5 µM HBL, HBL2 = 10 µM HBL, and HBL3= 15 µM HBL.

**Table 2 plants-12-03528-t002:** Effect of 28-homobrassinolide on chlorophyl *a*, chlorophyl *b*, total chlorophyl, carotenoids, metal tolerance index, and accumulation coefficient of *B. rapa* under lead stress.

Treatments	Chl *a* (mg g^−1^ FW)	Chl *b* (mg g^−1^ FW)	Total Chl (mg g^−1^ FW)	Carotenoids (mg g^−1^ FW)	Metal Tolerance Index	Accumulation Coefficient
C	0.588 ± 0.04 a	1.404 ± 0.04	9.96 ± 0.29 b	0.34 ± 0.3 d	-	-
Pb	0.311 ± 0.05 b	0.455 ± 0.1 a	3.829 ± 0.4 a	0.31 ± 0.2 a	28.1 ± 0.02 a	6.33 ± 0.02 a
HBL1	0.93 ± 0.06 b	2.197 ± 0.09 c	15.65 ± 0.19 c	0.45 ± 0.2 b	-	-
HBL2	1.219 ± 0.05 c	3.88 ± 0.02 e	25.53 ± 0.1 e	0.78 ± 0.3 d	-	-
HBL3	1.11 ± 0.04 c	3.578 ± 0.05 e	23.43 ± 0.1 e	0.72 ± 0.3 d	-	-
HBL1 + Pb	0.53 ± 0.05 b	1.837 ± 0.09 c	12.79 ± 0.4 c	0.71 ± 0.3 d	94.8 ± 0.09 b	3.69 ± 0.02 b
HBL2 + Pb	0.86 ± 0.012 b	2.794 ± 0.06 d	18.26 ± 0.7 d	0.79 ± 0.4 d	141 ± 0.12 d	4.007 ± 0.02 c
HBL3 + Pb	0.792 ± 0.04 b	2.241 ± 0.1 c	15.924 ± 0.6 c	0.76 ± 0.4 d	124 ± 0.98 c	3.17 ± 0.01 a

Data exhibit means ± SE of 4 replicates. Non-identical letters indicate significant differences between the treatments at *p* ≤ 0.05. C = control, Pb = 300 mg kg^−1^ Pb, HBL1 = 5 µM HBL, HBL2 = 10 µM HBL, and HBL3 = 15 µM HBL.

**Table 3 plants-12-03528-t003:** Effect of 28-homobrassinolide and lead on nutrition content and DPPH activity of *B. rapa* under lead stress.

Treatments	Mg^+2^ (mg g^−1^ DW)	Zn^+2^ (mg g^−1^ DW)	Na^+^ (mg g^−1^ DW)	K^+^ (mg g^−1^ DW)	DPPH Activity	Pb Uptake in Plant (mg/g)
C	6.72 ± 0.5 a	0.402 ± 0.02 b	437 ± 0.6 c	29.5 ± 0.35 c	45.9 ± 1.04	-
Pb	5.79 ± 0.02 a	0.237 ± 0.02 a	397.2 ± 0.55 a	19.85 ± 0.05 a	37.4 ± 1.2	0.143 ± 0.01 c
HBL1	8.52 ± 0.02 de	0.33 ± 0.02 ab	437 ± 0.7 c	29.6 ± 0.03 c	63.5 ± 0.9	-
HBL2	8.87 ± 0.02 e	0.47 ± 0.02 b	627.2 ± 0.7 g	41.2 ± 0.2 f	71.4 ± 0.91	-
HBL3	8.79 ± 0.02 e	0.36 ± 0.02 ab	517 ± 0.9 f	34.52 ± 0.03 e	65.6 ± 1.8	-
HBL1 + Pb	7.92 ± 0.02 b	0.31 ± 0.02 ab	428 ± 0.9 b	28.12 ± 0.01 b	55.5 ± 0.7	0.11 ± 0.2 b
HBL2 + Pb	8.5 ± 0.02 c	0.33 ± 0.02 ab	478 ± 0.6 e	31.76 ± 0.09 c	61.5 ± 0.78	0.05 ± 0.07 a
HBL3 + Pb	8.29 ± 0.02 bc	0.32 ± 0.02 ab	470 ± 0.89 d	28.7 ± 0.03 bc	57.8 ± 0.79	0.09 ± 0.04 b

Data exhibit means ± SE of 4 replicates. Non-identical letters indicate significant differences between the treatments at *p* ≤ 0.05. C = control, Pb = 300 mg kg^−1^ Pb, HBL1 = 5 µM HBL, HBL2 = 10 µM HBL, and HBL3 = 15 µM HBL.

## Data Availability

The original contributions presented in the study are included in the article, further inquiries can be directed to the corresponding authors.

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
