# Peer review of "28-Homobrassinolide Primed Seed Improved Lead Stress Tolerance in Brassica rapa L. through Modulation of Physio-Biochemical Attributes and Nutrient Uptake"

_plants, 2023, doi:10.3390/plants12203528_

Round 1
Reviewer 1 Report
Some problems need to be revised in the manuscript listed as follow:
1 In 2. Material and methods, the author didn’t describe how to arrange the germination test, no information on the test conditions. Also, the sampling time should be showed in each method.
2 In 2. Material and methods, the author didn’t provide the information for seed moisture content before and after priming. The difference of seed moisture content between seed samples maybe cause the changing of seed germinability. So, the author should maintain the moisture content at certain level during the priming.
3 in the Figure 4, what’s meaning for V2-V9? Also, the number of figure should be in sequence. Figure 5 should be marked after the Figure 4. Similar problem also presented in the Figure 3.
Author Response
Reviewer # 1
Some problems need to be revised in the manuscript listed as follow:
Comment 1: Material and methods, the author didn’t describe how to arrange the germination test, no information on the test conditions. Also, the sampling time should be showed in each method.
Response: The authors really appreciate the reviewer for the betterment of manuscript. Germination percentage was calculated using the formula: GP = seeds germinated/total seeds x 100. Results of germination also incorporated in revised manuscript. Under Pb stress, germination percentages were reduced. HBL, on the other hand, displayed a remarkable growth rate in table 1. Turnip plants exposed to Pb stress showed a 38.89% decrease in germination percentage compared to the control group. Turnip seeds fed with HBL2 demonstrated an increase in germination percentage of 5.5% when compared to control. Turnip seeds under Pb stress supplemented with HBL2 developing plants exhibited an increase of 43% compared to Pb only treatment.
Comment 2: Material and methods, the author didn’t provide the information for seed moisture content before and after priming. The difference of seed moisture content between seed samples maybe cause the changing of seed germinability. So, the author should maintain the moisture content at certain level during the priming.
Response:
The authors really appreciate the reviewer for the betterment of manuscript. Required information incorporated as per the kind suggestion of the reviewer. Primed seeds were properly rinsed before being placed on blotting paper for dehydration at room temperature for 3 hours to retain the original moisture content.
Comment 3: in the Figure 4, what’s meaning for V2-V9? Also, the number of figure should be in sequence. Figure 5 should be marked after the Figure 4. Similar problem also presented in the Figure 3.
Response:
Thank you for your keen interest in the improvement of the manuscript. V1 – V8 represents the 8 treatments such as V1=Control, V2=Pb=300 mgKg-1, V3=1 µM, V4=5 µM, V5=10 µM, V6=HBL1+Pb, V7=HBL2+Pb, V8=HBL3+Pb Figure 3, 4, and 5 marked as per suggested by reviewer.
Reviewer 2 Report
Issues related to the study of stress reactions in plants, are very important. First, the development of ideas about the responses of plants to the impact of adverse environmental conditions is of scientific interest and makes it possible to better understand the patterns of plant functioning. Secondly, this area of plant physiology is of applied importance, since the identification of the mechanisms of resistance and adaptation of plants to adverse environmental factors opens up broad prospects for the development of plant breeding and biotechnologies. In this regard, the research topic of Mawra Khan et al. is important and relevant.
The applied methods are adequate and modern. In general, the results are clear and objective. However, there are some remarks that need to be noted.
1. The article submitted for review lines are not numbered, which makes reviewing difficult.
2. Based on the results obtained by the author, the concentration of 28-homobrassinolide 2 is the most effective (both without the addition of lead and under stress). But the authors do not note this anywhere.
3. In my opinion, good results have been obtained in relieving the stress response with 28-homobrassinolide. The conclusions lack recommendations for using the results obtained in practical activities (for example, in agriculture).
Based on the mentioned above, I think that this article can by recommended for publication in the «Plants» after revision.
Author Response
Reviewer #2
Issues related to the study of stress reactions in plants, are very important. First, the development of ideas about the responses of plants to the impact of adverse environmental conditions is of scientific interest and makes it possible to better understand the patterns of plant functioning. Secondly, this area of plant physiology is of applied importance, since the identification of the mechanisms of resistance and adaptation of plants to adverse environmental factors opens up broad prospects for the development of plant breeding and biotechnologies. In this regard, the research topic of Mawra Khan et al. is important and relevant.
The applied methods are adequate and modern. In general, the results are clear and objective. However, there are some remarks that need to be noted.
Comment 1. The article submitted for review lines are not numbered, which makes reviewing difficult.
Response:
Authors feel very sorry for inconvenience. Correction incorporated in revise manuscript.
Comment 2. Based on the results obtained by the author, the concentration of 28-homobrassinolide 2 is the most effective (both without the addition of lead and under stress). But the authors do not note this anywhere.
Response:
Thank you for your remarkable recommendation. The results regarding 28-homobrassinolide concentration 2 incorporated relative results section.
Comment 3. In my opinion, good results have been obtained in relieving the stress response with 28-homobrassinolide. The conclusions lack recommendations for using the results obtained in practical activities (for example, in agriculture).
Response:
Thank you for your valuable suggestion. Corrections incorporated regarding future recommendations in revised manuscript in conclusion section.
Based on the mentioned above, I think that this article can by recommended for publication in the «Plants» after revision.

Reviewer 3 Report
Paper "28-homobrassinolide primed seed improved lead-stress tolerance in Brassica rapa L. through modulation of physio-biochemical attributes and nutrient uptake" is a well-prepared, consistent manuscript, with varied investigation methods and consistent results.
The paper analyzes in detail the influence of 28-homobrassinosteroids on the improvement of the reaction mode of Brassica rapa plants, grown on lead-polluted soil. The results obtained are consistent, the vast majority of the investigated parameters showing variations in the same direction. The protective effect of 28-homobrassinosteroids is demonstrated by quantitative investigations on growth processes, as well as by determining various biochemical and physiological parameters.
The degree of novelty of the problem addressed by the authors is moderate, because the protective effect of this compound against the stress induced by the presence of heavy metals (in particular) but also that induced by drought, salinity, pathogens, has already been studied by various authors in many plant species, with similar results.
Some changes and additional explanations are requested, to improve the quality of the manuscript.
1. Introduction:
- "environmental contamination with various contaminants" - repetition, please rephrase;
- "Heavy metals accumulated in soil in minute amounts" - maybe in minimal amounts.
2. Material and methods: "Thinning was done when the plants were mature enough.." please mention how many days "mature enough" means.
Please elaborate on the statement from chapter 2.1: "Shoot length, leaf area, root length, and leaf number per plant were all measured.": how was the leaf area measured? With what instrument were the lengths of shoots and roots measured?
Note: although the area of the leaf appears to have been measured in this chapter, I did not find in Chapter 3 Results given about this parameter. Please clarify this aspect.
Table 1 - All figures and tables must be self-explanatory. That is why the abbreviations must be mentioned: ex Shoot FW - Shoot fresh weight (the observation is valid for all other tables and figures).
Chapter 4 Discussions: ”. Many ions are known to be greatly prevented from entering their absorption sites on the roots by lead" - something is missing in the expression, please rephrase.
- in the Instructions for authors from the journal Plants, references from the Bibliography must be mentioned in the text by numbers, in the order of appearance. Please review the entire manuscript in this regard.
The References are not written according to the journal requirements (especially regarding the names of the journals and the names of the authors).
I had no problems understanding the text.
Author Response
Reviwer#3
Paper "28-homobrassinolide primed seed improved lead-stress tolerance in Brassica rapa L. through modulation of physio-biochemical attributes and nutrient uptake" is a well-prepared, consistent manuscript, with varied investigation methods and consistent results.
The paper analyzes in detail the influence of 28-homobrassinosteroids on the improvement of the reaction mode of Brassica rapa plants, grown on lead-polluted soil. The results obtained are consistent, the vast majority of the investigated parameters showing variations in the same direction. The protective effect of 28-homobrassinosteroids is demonstrated by quantitative investigations on growth processes, as well as by determining various biochemical and physiological parameters.
The degree of novelty of the problem addressed by the authors is moderate, because the protective effect of this compound against the stress induced by the presence of heavy metals (in particular) but also that induced by drought, salinity, pathogens, has already been studied by various authors in many plant species, with similar results.
Some changes and additional explanations are requested, to improve the quality of the manuscript.
Comment: 1. Introduction: "environmental contamination with various contaminants" - repetition, please rephrase;
Response: Thank you for suggestion. Correction incorporated in revise manuscript.
Comment:- "Heavy metals accumulated in soil in minute amounts" - maybe in minimal amounts.
Response: Thank you for suggestion. Correction incorporated in revise manuscript.
Comment: 2. Material and methods: "Thinning was done when the plants were mature enough.." please mention how many days "mature enough" means.
Response: Thank you for keen interest in improvement of manuscript. Thinning was done when the plants were mature enough about 25 days after initiation of germination, and two seedlings were kept in each assigned pot. Germination percentage was calculated by using the formula; Germinated seeds/Total no. of seeds ×100. Destructive harvesting was carried after 48 days after germination.
Comment: Please elaborate on the statement from chapter 2.1: "Shoot length, leaf area, root length, and leaf number per plant were all measured.": how was the leaf area measured? With what instrument were the lengths of shoots and roots measured?
Response: . Shoot length, leaf area, root length, and leaf number per plant were all measured. The length of root and shoot was measured by using ruler. Whereas, leaf area is removed from 2.1.
Comment: Note: although the area of the leaf appears to have been measured in this chapter, I did not find in Chapter 3 Results given about this parameter. Please clarify this aspect.
Response: Thank you for your concern. Leaf area from methodology and results section removed in revised manuscript.
Comment: Table 1 - All figures and tables must be self-explanatory. That is why the abbreviations must be mentioned: ex Shoot FW - Shoot fresh weight (the observation is valid for all other tables and figures).
Response:
Thank you for suggestion. Correction incorporated in all figures and tables.
Comment: Chapter 4 Discussions: ”. Many ions are known to be greatly prevented from entering their absorption sites on the roots by lead" - something is missing in the expression, please rephrase.
Response: Thank you for suggestion. Lead is known to block several ions from entering their absorption sites on the roots (Godbold and Kettner, 1991),
Comment: - in the Instructions for authors from the journal Plants, references from the Bibliography must be mentioned in the text by numbers, in the order of appearance. Please review the entire manuscript in this regard.
Response:
Thank you for suggestion. Correction incorporated in revised manuscript.
Comment: The References are not written according to the journal requirements (especially regarding the names of the journals and the names of the authors).
Response:
Thank you for suggestion. Correction incorporated in references according to journal instruction.
